# Analysis of individual differences in pelvic and spine alignment in seated posture and impact on the seatbelt kinematics using human body model

**Norihiro Nishida** [1]*, **Tomohiro Izumiyama** [2], **Ryusuke Asahi** [2], **Fei Jiang** [3], **Junji Ohgi** [3], **Hiroki Yamagata** [1], **Yasuaki Imajo** [1], **Hidenori Suzuki** [1], **Masahiro Funaba** [1], **Shigeru Sugimoto** [2], **Masanobu Fukushima** [2], **Xian Chen** [3], **Takashi Sakai** [1]

1 Department of Orthopedic Surgery, Yamaguchi University Graduate School of Medicine, Ube City, Yamaguchi Prefecture, Japan, 2 Crash Safety Development Department, Vehicle Development Division, Mazda Motor Corporation, Aki-gun, Hiroshima Prefecture, Japan, 3 Faculty of Engineering, Yamaguchi University, Ube City, Yamaguchi Prefecture, Japan

* nishida3@yamaguchi-u.ac.jp

**Data Availability Statement:** All relevant data are within the manuscript and S1 Data.

## Abstract

Analysis using human body models has been performed to reduce the impact of accidents; however, no analysis has shown a relationship between lumbar and pelvic/spine angle and seat belts in reducing human damage from accidents. Lumbar and pelvic/spine angles were measured in 75 individuals and the measurements were used to create three different angles for the Total Human Model for Safety model. In the present study, we focused on lumber lordosis (LL) and pelvic angle (PA). A normal distribution and histogram were used for analysis of PA (01, 10, and 50). The Total Human Model for Safety, including LL and PA, was corrected using finite element software. Simulations were conducted under the conditions of the Japan New Car Assessment Programme (JNCAP) 56 kph full lap frontal impact. Using the results of the FEM, the amount of lap-belt cranial sliding-up, anterior movement of the pelvis, posterior tilt of the pelvis, head injury criterion (HIC), second cervical vertebrae (C2) compressive load, C2 moment, chest deflectiou (upper, middle, and lower), left and right femur load, and shoulder belt force were measured. The lap-belt cranial sliding-up was 1.91 and 2.37 for PA10 and PA01, respectively, compared to PA50; the anterior movement of the pelvis was 1.08 and 1.12 for PA10 and PA01, respectively; and the posterior tilt of the pelvis was 1.1 and 1.18 for PA10 and PA01, respectively. HIC was 1.13 for PA10 and 1.58 for PA01; there was no difference in C2 compressive load by PA, but C2 moment increased to 1.59 for PA10 and 2.72 for PA01. It was found that as LL increases and the PA decreases, the seat belt becomes likely to catch the iliac bone, making it harder to cause injury. This study could help to reconsider the safe seat and seatbelt position in the future.

**Funding:** Mazda Motor Corporation provided $30,000 to cover joint 2017 research expenses for this study. We have financial relationships to disclose. However, The funder provided support in the research materials (car seat, FEM software, English editing fee, OASYS, and THUMS), but did not have any additional role in the study design, data collection and analysis, decision to publish, or preparation of the manuscript. Norihiro Nishida developed research plan. Norihiro Nishida and Hiroki Yamagata measured X-ray data. Norihiro Nishida, Fei Jiang, and Xian Chen analysed FEM using THUMS. Junji Ohgi, Hidenori Suzuki, Yasuaki Imajo, Masahiro Funaba, and Takashi Sakai played role of guidance and modification of reseach. Tomohiro Izumiyama, Ryusuke Asahi, Shigeru Sugimoto, and Masanobu Fukushima are not have any additional role in the study design, data collection and analysis, decision to publish, or preparation of the manuscript.

**Competing interests:** Mazda Motor Corporation provided $30,000 to cover joint 2017 research expenses for this study. This does not alter our adherence to PLOS ONE policies on sharing data and materials.

# Introduction

Although the global death rate from motor vehicle accidents is declining, approximately 1.35 million people are killed each year as a result of road traffic crashes according to the World Health Organization's Road Traffic Crashes database. Whiplash injury can reduce an accident victim's quality of life for a significant amount of time, ranging from weeks to a few months, with symptoms persisting for years in some cases [1]. Various efforts have been made to reduce human injuries from car accidents by improving car seat belts and the car itself, including the materials used in manufacture, as well as by analyzing car accidents. Seat belts have proven to be important in reducing injuries to car occupants during crashes [2]. However, deaths and injuries from car accidents are still too frequent and there remains considerable scope for improving car seats and seat belts, especially since there are clinical reports of injury resulting from seat belts during accidents [3–5]. These studies do not refer to the mechanism by which seat belts, which are meant to protect humans, can also cause injuries in some circumstances. Actual crash studies using humans are not practical or ethically feasible. Crash analysis using dummy dolls in cars is important, but this is quite expensive and can't be done more than once [6]. Computer simulation using the finite element method (FEM) is an important way to analyze injury from car accidents, but so far there are few published reports. Kimpira analyzed injuries from car accidents and used FEM human body models to simulate the safety profile of various seats and seat belts (HBMs) [7].

They did this by adapting a previously used male FEM model to a female FEM model. In addition, Danelson *et al.* performed a pediatric-sized FEM and brain injury analysis for a car crash [8]. Davis *et al.* performed a seated FEM analysis and reported that pelvic fractures were more likely to occur in female models [9].

However, it is difficult to fully replicate injuries from car accidents. To make the analysis more thorough, we considered the possibility that the occupant's seating alignment was also involved. As a basis for this hypothesis, it is well known that changing from a standing to a sitting position causes changes in lumbar and pelvic alignment [10–17]. Individual differences in skeletal and seated alignment may explain why simulations do not fully replicate accidents.

In the present study we investigated factors that influenced FEM simulation and the effect of angle of the spine and pelvis on the injury to a person in an accident when sitting in a car seat.

THUMS version 4.0 (THUMS ver. 4.0; TOYOTA MOTOR CORPORATION, Toyota, Aichi, Japan) and the finite element method (FEM) were used to analyze the impact to seat belts and the human body at the time of a collision by modeling the skeletal radiological data of a person sitting in a car seat. A better understanding of human and seat belt movement according to the angle of the spine and pelvis should lead to improved analysis of human injury mechanisms and risk reduction.

# Material and methods

This study was conducted in collaboration with project partner Mazda Motor Corporation (Aki-gun, Hiroshima, Japan). The research was approved by the ethics committee at the Center for Clinical Research, Yamaguchi University Hospital (Ube, Japan; approval no. H29-052 and 2021–053). Written informed consent for this study and its publication was obtained from all subjects. And, the individual (for example Fig 1) in this manuscript has given written informed consent (as outlined in PLOS consent form) to publish these case details.

To measure lumbar and pelvic alignment in the sitting position, seventy-five adult participants (45 men and 30 women) with no history of spine disease, lower limb surgery, or current lower back or leg pain were included in this study. Their mean age was 45.8 years (range, 24–

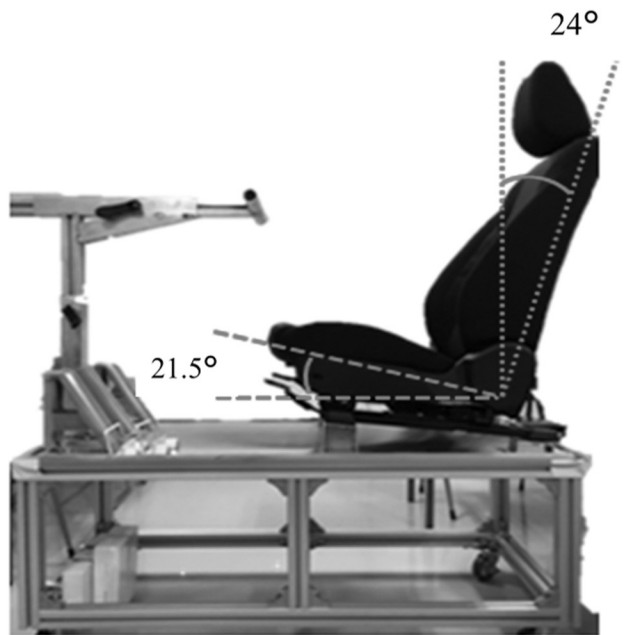
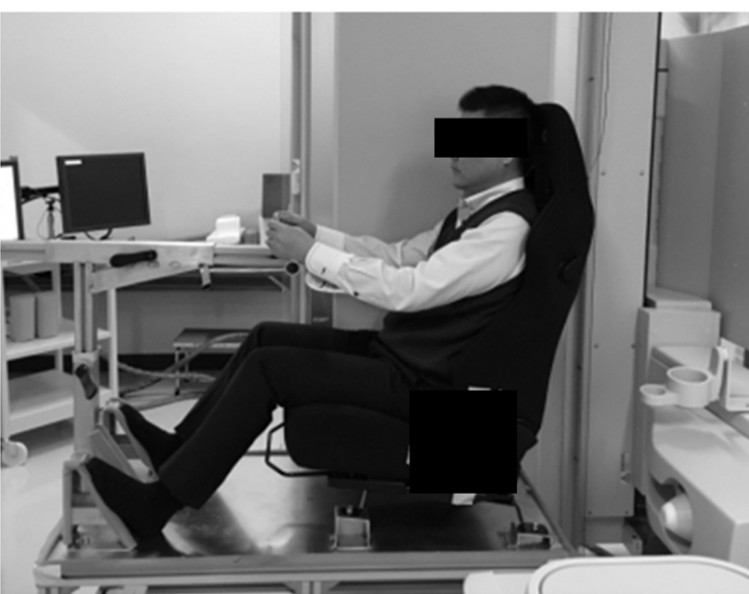

**Fig 1.** The positions taken by study participants for seat (*left*) and sitting position (*right*).

70 years), their mean height was 164.8 cm (range, 149–187 cm), and their mean body mass index (BMI) was 24.0 kg/m2 (range, 17.5–42.6 kg/m2).

Lateral x-ray images of the global spine alignment in the sitting position were acquired with participants in a car seat (same as the driver's side seat of Mazda3). The car seat size was designed to fit in the radiography room, and the seat height was adjusted to allow lateral x-ray imaging of the spine and pelvis, with the steering wheel and pedals positioned similarly to those in a Mazda 3 vehicle. The seat's surface and reclining angles were fixed, but slight back-and-forth adjustments were possible (Fig 1). To limit x-ray exposure, imaging was performed only once with the car seat in a fixed position.

In the present study, we focused on lumbar lordosis (LL), the sacral slope (SS), and pelvic angle (PA). After imaging of the spine, the following angles were measured in the sitting position: lumbar lordosis (LL) between the superior end plate of the 1st to the superior end plate of the 5th lumbar vertebrae; the sacral slope (SS) between sacrum and the horizontal line; pelvic angle (PA) between the line from the pubic symphysis to the anterior superior iliac spine (ASIS) and perpendicular line were measured [18–20] (Fig 2). This data were reflected in the FE analysis models.

## FE analysis

The human model for collision analysis, THUMS ver. 4.0 AM50, was used. Alignment of THUMS was modified using the pre-processing software Oasys PRIMER™ (ARUP, London, UK). Both software packages were purchased.

Since the analysis of each subject's data was time consuming, we focused on PA. In order to select representative PA values, a normal distribution and histogram were used for PA analysis (Fig 3). The results showed that the 1st percentile of the PA was 67˚, the 10th percentile was 57˚, and the 50th percentile was 45˚. The LL angles of the volunteers for each PA angle were -8˚, -8˚, and 27˚, while the SS angles for each PA angle were -20˚, -10˚, and 5˚. To create a model corresponding to those angles, skeletal alignment in the THUMS model was modified

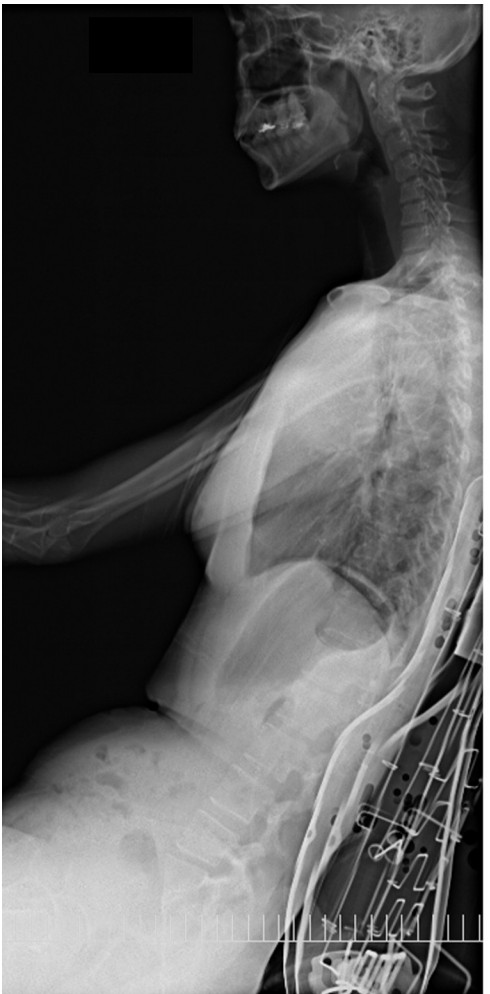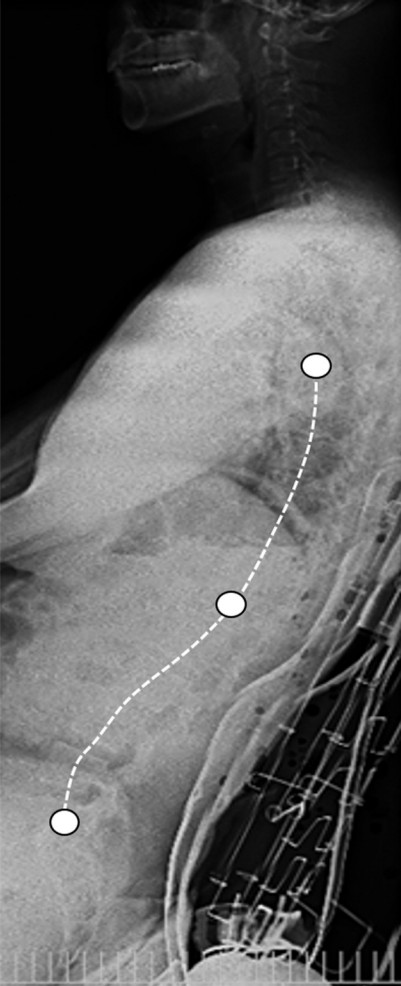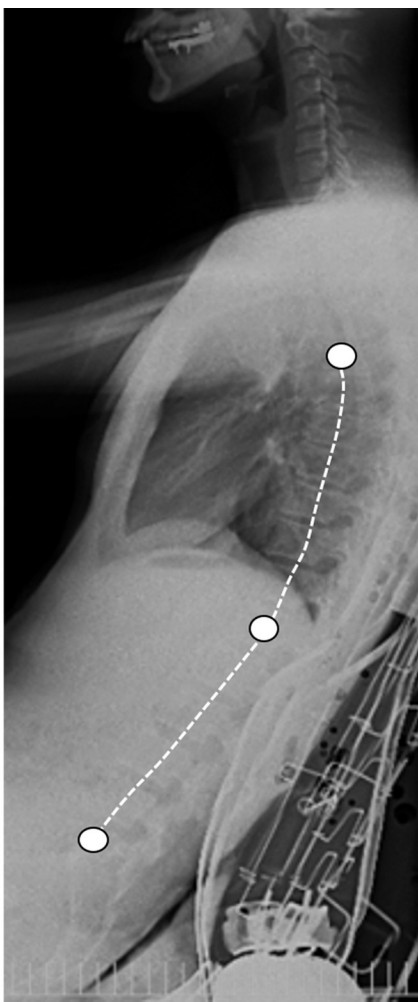

**Fig 2.** Lateral radiographs of three volunteers (left). There were volunteers with a lumbar lordosis (center) and volunteers with decreasing lumbar lordosis and pelvic tilt (right).

by giving Prescribed-Motion (enforced displacement) to the vertebrae of the thoracic spine, lumbar spine and pelvis.

In this study, simulations were conducted under the conditions of the Japan New Car Assessment Programme (JNCAP) 56 kph full lap frontal impact. Car models, the position of HBMs, and restraint state by seat belt were the same as the simulation conditions. The shoulder belt was set at the chest and the lap belt was set to fit the pelvis (Fig 3). For certain conditions, below the wrist, below the knee, and the cervical spine of THUMS were restrained. The hip joint was not restrained for flexion and extension, but was restrained for abduction, adduction, and rotation (Fig 4).

Using the results of the FEM, the amount of lap-belt cranial sliding-up anterior movement of the pelvis, posterior tilt of the pelvis for seeing abdominal, aorta sand spine injury. Head injury criterion (HIC), second cervical vertebrae (C2) compressive load, C moment for seeing cervical spine injury, chest deflectiou (upper, middle, and lower) for seeing chest injury and rib injury, left and right femur load, and shoulder belt force for seeing limbs injury were measured. Analysis and animation creation were performed using Animator software (CDH-Japan

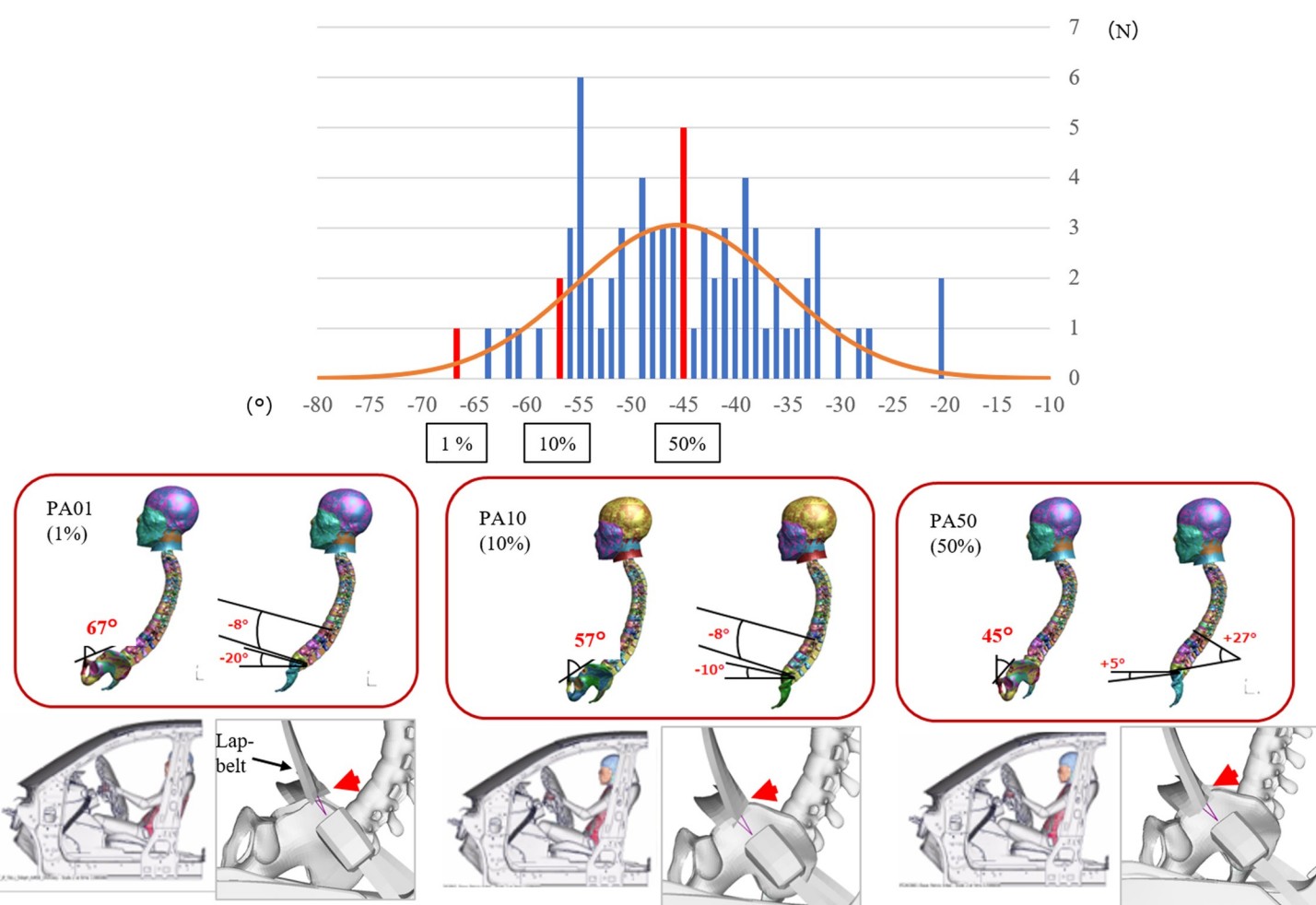

**Fig 3.** Normal distribution and histogram of PA (upper figure). Angle correction model of each model (middle figure). PA01; PA 67˚, LL -8˚, SS -20˚. PA10; PA 57˚, LL -8˚, SS -10˚. PA50; PA 45˚, LL 27˚, SS 5˚. Diagram of the seat of each model and the position of the lap-belt (below).

LTD., Yokohama, Kanagawa, Japan). Using the alignment, we measured the movement and rotation of the body and seatbelt during car crash and analyzed the injury risk.

## Results

### Radiological data

The mean values of alignment in the sitting position were as follows: LL, -0.49˚±9.58˚(range, -26˚ to 26.6˚); SS, -16.5˚±8.64˚ (range, -41˚ to 4.3˚); and PA, -45.57˚ ±9.80˚(range, -67.0˚ to -20.0˚).

### FEM

Fig 5 shows a whole human model of a collision from the riding situation and the extracted seat belts, vertebral pelvis, and head.

The measurements are shown in Table 1. At the time of the collision, all parameter had increased. Lap-belt cranial sliding-up was 10.8, 20.6, and 25.7mm, anterior movement of pelvis was 139.8, 154.0, and 165.1mm for PA50, PA10, and PA01, posterior tilt of the pelvis was 41.75, 45.05, and 46.78˚, HIC was 532.9, 601.4, and 840.9, C2 moment was 1.88, 2.99, 5.11mm,

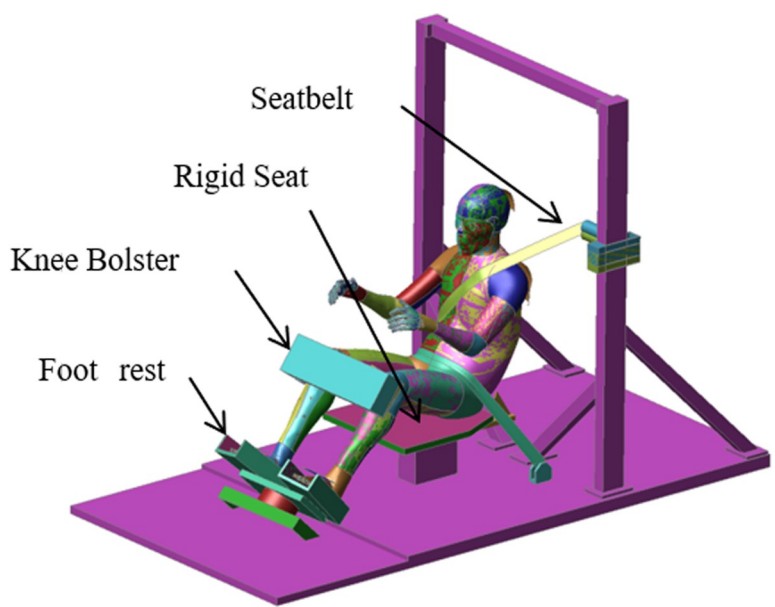

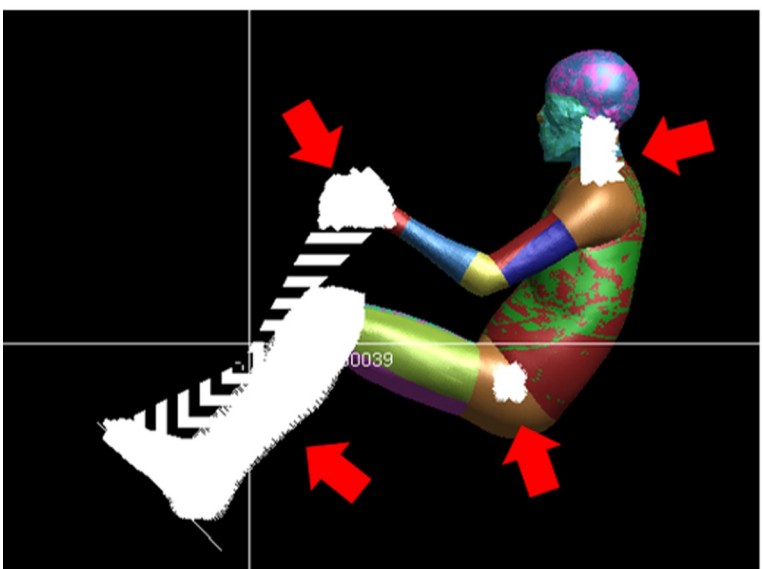

**Fig 4.** Initial posture of THUMS (above). Below the wrist, below the knee, and the cervical spine of THUMS were restrained (arrows). In addition, the hip joint was not restrained to flexion and extension; however, abduction, adduction, and rotation were restrained (see below).

and chest deflection lower was 25.4, 28.7 and 30.5 for PA50, PA10, and PA01. These parameter were gradually increased by from PA50 to PA01. The other parameters did not show a gradual increasing trend with PA.

C2 compressive load, chest deflectiou, and femur and shoulder was 1.66, 1.55, and 1.88N, The graph shows the ratio of damage to PA10 and PA01 when the damage to PA50 was set to 1. The lap-belt cranial sliding-up was 1.91 and 2.37 for PA10 and PA01, respectively, compared

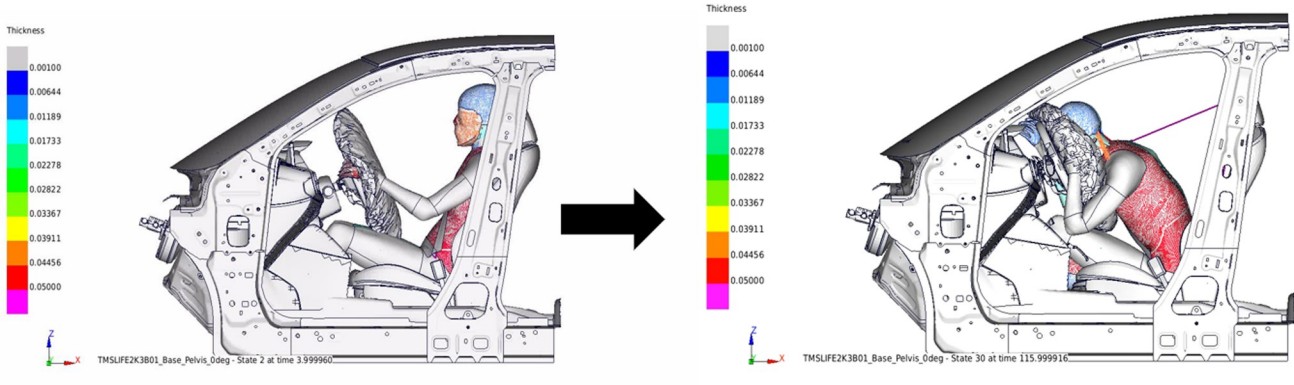

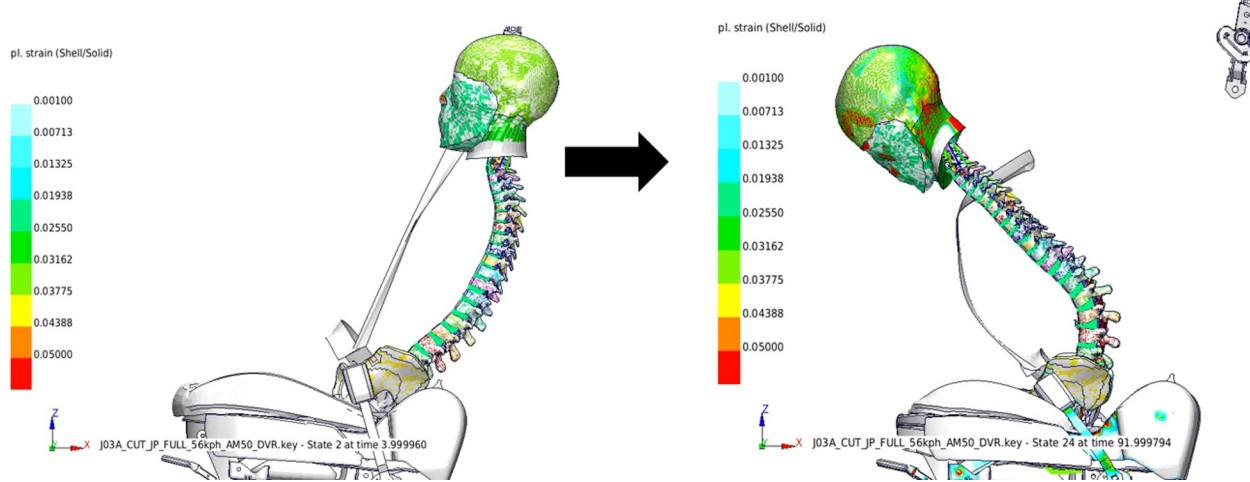

**Fig 5. Initial position of the collision analysis.** The upper two figures show the whole model and the lower figures shows only the spine and head. The left image shows the position before the experiment started and the right image shows the position after the collision. Overall, it shifted forward.

**Table 1. Measured values for each parameter.**

| Parameter | PA50 | PA10 | PA01 |
|---|---|---|---|
| Lap-belt cranial sliding-up (mm) | 10.8 | 20.6 | 25.7 |
| Anterior movement of pelvis(mm) | 139.8 | 154.0 | 165.1 |
| Posterior tilt of the pelvis (°) | 41.75 | 45.05 | 46.78 |
| HIC | 532.9 | 601.4 | 840.9 |
| C2 Compressive Load (N) | 1.66 | 1.55 | 1.88 |
| C2 Moment (mm) | 1.88 | 2.99 | 5.11 |
| Chest Deflectiou Upper (mm) | 35.4 | 37.5 | 31 |
| Chest Deflectiou Middle (mm) | 28.6 | 32.6 | 30.6 |
| Chest Deflectiou Lower (mm) | 25.4 | 28.7 | 30.5 |
| L Femur Load (N) | 0.83 | 0.81 | 1.0 |
| R Femur Load(N) | 1.62 | 1.31 | 1.65 |
| Shoulder Belt Force (N) | 5.57 | 5.52 | 5.66 |

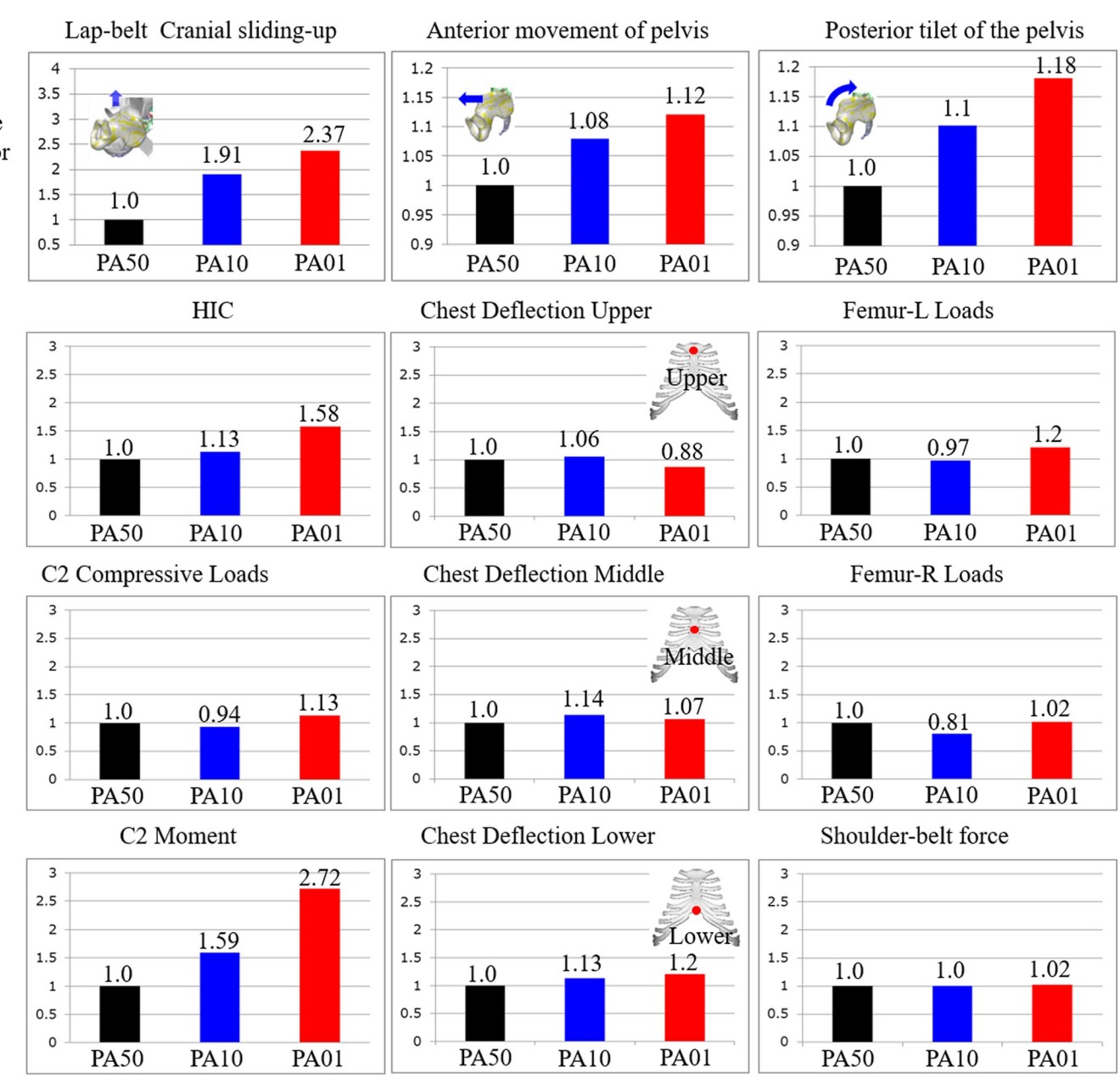

**Fig 6. The graph shows the ratio of damage to PA10 and PA01 when the damage to PA50 is set to 1.** The vertical axis is the ratio and the graph for each parameter is shown.

to PA50; the anterior movement of the pelvis was 1.08 and 1.12 for PA10 and PA01, respectively; and the posterior tilt of the pelvis was 1.1 and 1.18 for PA10 and PA01, respectively.

The HIC was 1.13 for PA10 and 1.58 for PA01; there was no difference in C2 compressive load by PA, but C2 moment increased to 1.59 for PA10 and 2.72 for PA01. There was no difference in the loading of the chest, femoral, and shoulder belts (Fig 6).

PA50 tended to have less Lap-belt cranial sliding-up, Anterior movement of pelvis, HIC, and C2 moment than PA10 and PA01. There were no characteristic differences in the chest or lower limbs due to alignment.

## Discussion

This study investigated alignment of the human body when sitting in a car seat; thus, simulating human damage patterns of accidents and investigating seat belt and seat positions. One of the aims was to analyze car crashes using a human model and help decrease the risk of injury during an accident. The study also provides evidence for the development of other vehicle seats and human simulation analysis.

Several computer simulation analyses using THUMS and FEM for automobile accidents have been reported. Kimpara et al. analyzed a female model and compared the results with a male model [21]. Andersson et al. performed an analysis on child size and reported that light vehicles required higher performance interior restraint systems than larger vehicles [22]. Danelson et al. and Digges et al. analyzed lung contusions during motor vehicle accidents [23, 24]. Klein et al. reported that body shape altered impact to the femur [25]. Mattos et al. analyzed the damage to the head and cervical spine during a rollover [26]. Jones et al. performed an analysis of lumbar spine fractures at the time of the collision [27]. Paas et al. analyzed the mechanism of shoulder injury at the time of an accident [28]. Xiao et al investigated of chest injury mechanism of seat belt using FEM [29]. Our analysis also showed movement of shoulder, chest, and cervical spine, as previously reported. These analyses can be viewed as useful research for the development of human models and improving vehicle safety. However, there are challenges resulting from collision conditions and model validation, and there was no analysis focusing on vertebral angle.

There were some reports of individual differences in the angle of lumbar and pelvis with sitting position. Studies by Suzuki et al. and Endo et al. examined the alignment in standing and sitting positions in middle-aged and elderly individuals, and in young adults, respectively [12, 13]. Lee et al. reported that lumbar spine lordosis and pelvic tilt were decreased when study participants were sitting [14]. Nishida et al. reported that body alignment changes with the car seat [17]. However, there were no reports examining the effects of lumbar spine or pelvic alignment in the analysis of car accidents. In the present analysis, three models were created with different angles of the pelvis and lumbar vertebrae, and it was found that there were large individual differences in the angle changes. In the present analysis, we focused on these clinical reports.

There have been several reports on clinical injuries caused by traffic accidents. It has been clinically identified that there is an increased whiplash injury risk when the occupants head is rotated or inclined prior to the impact [30, 31]. Radanov et al reported a higher incidence of rotated or inclined head position at the time of impact was correlated with increased severity of symptoms such as neck pain, shoulder pain, and headache [32]. In the present analysis, C2 compressive load also increased during the crash, and HIC and the C2 moment was exacerbated by the larger PA. The present analysis also showed that the accident added movement to head and the cervical spine.

Herath reported a shearing transection of a gastroduodenal junction caused by an inappropriate seat belt [33]. Muraoka et al reported that uterine trauma and Intrauterine fetal death caused by seatbelt injury [34]. Tomic et al reported that seat-belt abdominal aortic injury and rib fracture [35]. Ramachandra et al reported on seat belt-induced increase in abdominal pressure during an accident [36]. Abbas et al. reported that seatbelt-related injuries include spinal, abdominal or pelvic injuries and the presence of a seatbelt sign must raise the suspicion of an intra-abdominal injury due to seatbelt repositioning during traffic accidents. The presence of a seatbelt sign must raise the suspicion of an intra-abdominal injury [37]. In the present study, actual collision analysis was carried out in three human models based on the measured data of spine and pelvis angles. Lap-belt cranial slide-up, anterior movement of the pelvis, and

posterior tilt of the pelvis moved in all alignment conditions, suggesting that traffic accidents can cause abdominal injuries, aortic injuries, and spinal fractures. Some researcher said inappropriate seat belt use, but we found that increasing LL and decreasing PA allowed for better seat belt application to the iliac bone and better control of human movement. It has the potential to reduce injury in the event of an accident. Jiang et al. reported a higher risk of seat belt injury when the seat belt was above the ASIS, and the present study also found that as pelvis tilt decreases, the seat belt position shifted to the head side and the ability to control the body decreased [38]. This study found that the spinal alignment, pelvic tilt and seatbelt position may decrease/increase the risk of injury.

There are several limitations to our study. The size and imaging range of the x-ray equipment did not allow frontal imaging in the sitting position and only allowed lateral imaging at one angle (not adjustable by reclining). In addition, we studied only one tissue strength of bones and organs in THUMS; therefore, age-related changes, such as osteoporosis, were not taken into account. In addition, the soft tissue thickness of the thighs and abdomen, and variations in the size of the pelvis were not included in this model. With computer simulation, the results are the same regardless of how many times the experiment is repeated and hence statistical analysis was not performed. These are issues to be addressed in the future.

Despite these limitations, an analysis of a hypothetical car accident using a human model with altered vertebral angle and PA showed changes in seat belt position and damage to the human model. This will be a useful reference for future accident analysis and research on the appropriate seat shape and seat belt position.

## Conclusions

We measured the angle of the lumber and pelvis when sitting in the car seat using real volunteers and used the data to perform a frontal crash analysis in THUMS. It was found that as LL increases and PA decreases, the seat belt becomes likely to catch the iliac bone, reducing the risk of human injury. Many additional models would have to be analyzed to fully understand the importance of, and interactions between, skeletal alignment parameters. However, this study could contribute to improving the safety of seats and seat belt positioning in the future.

## Supporting information

**S1 Data.**
(XLSX)

## Acknowledgments

We thank Masaki Ueno of Mazda Motor Corporation for suggesting that we work together on joint research.

Medical English Service (Kyoto, Japan) provided professional English-language editing of the manuscript for this article.

## Author Contributions

**Conceptualization:** Norihiro Nishida, Yasuaki Imajo.

**Data curation:** Hiroki Yamagata, Masahiro Funaba.

**Formal analysis:** Norihiro Nishida, Hidenori Suzuki.

**Funding acquisition:** Norihiro Nishida, Ryusuke Asahi, Masanobu Fukushima.

**Investigation:** Tomohiro Izumiyama.

**Project administration:** Hiroki Yamagata.

**Software:** Fei Jiang.

**Supervision:** Junji Ohgi.

**Validation:** Tomohiro Izumiyama, Ryusuke Asahi, Fei Jiang, Shigeru Sugimoto.

**Writing – original draft:** Norihiro Nishida.

**Writing – review & editing:** Xian Chen, Takashi Sakai.

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
