## [Decision Letter · Decision Letter 0]

7 Apr 2021

PONE-D-20-40025

Analysis of Individual Differences in Pelvic and spine alignment in seated posture and Impact on the Seatbelt Kinematics using human body model

PLOS ONE

Dear Dr. Nishida,

Thank you for submitting your manuscript to PLOS ONE. After careful consideration, we feel that it has merit but does not fully meet PLOS ONE’s publication criteria as it currently stands. Therefore, we invite you to submit a revised version of the manuscript that addresses the points raised during the review process.

We look forward to receiving your revised manuscript.

Kind regards,

YunJu Lee

Academic Editor

PLOS ONE

Journal Requirements:

2. Thank you for providing a statement in your Methods section that informed written consent was given by all participants. We ask that you additionally include this statement in the Ethics Statement.

Please amend your Methods section to clarify where the THUMS and OASYS software products can be accessed, and if they are freely available to the public.

3. Thank you for including your ethics statement:  "This study was conducted in collaboration with project partner Mazda Motor Corporation (Aki-gun, Hiroshima, Japan) with approval from the ethical review board of the corresponding author’s host institute.".   

5. Thank you for stating the following in the Financial Disclosure section:

'Mazda Motor Corporation provided $30,000 to cover joint 2017 research expenses for this study.'

We note that one or more of the authors have an affiliation to the commercial funders of this research study: Mazda Motor Corporation

6. Please amend either the title on the online submission form (via Edit Submission) or the title in the manuscript so that they are identical.

7. We note that Figure 1 includes an image of a participant in the study. 

As per the PLOS ONE policy (http://journals.plos.org/plosone/s/submission-guidelines#loc-human-subjects-research) on papers that include identifying, or potentially identifying, information, the individual(s) or parent(s)/guardian(s) must be informed of the terms of the PLOS open-access (CC-BY) license and provide specific permission for publication of these details under the terms of this license.

Please download the Consent Form for Publication in a PLOS Journal (http://journals.plos.org/plosone/s/file?id=8ce6/plos-consent-form-english.pdf). The signed consent form should not be submitted with the manuscript, but should be securely filed in the individual's case notes.

Please amend the methods section and ethics statement of the manuscript to explicitly state that the patient/participant has provided consent for publication: “The individual in this manuscript has given written informed consent (as outlined in PLOS consent form) to publish these case details”.

Reviewers' comments:

Reviewer's Responses to Questions

**Comments to the Author**

1. Is the manuscript technically sound, and do the data support the conclusions?

Reviewer #1: Yes

Reviewer #2: Partly

2. Has the statistical analysis been performed appropriately and rigorously? 

Reviewer #1: Yes

Reviewer #2: No

3. Have the authors made all data underlying the findings in their manuscript fully available?

Reviewer #1: Yes

Reviewer #2: Yes

4. Is the manuscript presented in an intelligible fashion and written in standard English?

Reviewer #1: Yes

Reviewer #2: Yes

5. Review Comments to the Author

Reviewer #1: This study measured the angle between lumber and pelvis from volunteers and performed frontal crash analysis with the THUMS model. This study will help the future accident analysis, such as seat and seat belt development. This article should be of interest to the reader of PLOS ONE, especially those in the vehicle safety. Some more detailed information, as well as an editorial correction, are needed. Following comments are offered to the authors:

1. Line 95: Should it be PA (01, 10, and 50)?

2. Line 120: Please show more details for the previous study. For example, what variables were involved in Kimpira’s study? How different is the predicted injury than the real injury?

3. Line 134-137: This paragraph should be in the Acknowledgement

4. Line 144: Are the seat size and seat height similar to the ones in the Mazda car?

5. Line 183: For all the results, please also provide standard deviations. Although range is useful, standard deviation is considered the more reliable measure for statistical analysis.

6. Line 191/Table 1: Please explain the parameters better. Will the change of these parameters result in injury? Can you calculate the injury risk? One aim of this study is to help decrease the risk of injury during an accident. Therefore, the estimated injury probability will be helpful.

Reviewer #2: 1. The introduction is too short. The authors need to identify the contribution, novelty, and related works comprehensively.

2. Methods- “The seat’s surface and reclining angles were fixed.” Due to the X-ray was used to perform once to collect the data, how to ensure the sitting posture is correct or the same as usual when the participants driving? Does the reclining angle of the seat affect the lumbar angle and pelvis?

3. The FE analysis needs emphasis clearly. How to measure each measure (LL, SS…)? Not only provided the definitions but also the detail of collection and measurements.

4. Lacks of statistical analysis results to support the findings.

5. The discussion is weak. Authors need to focus on the results of the study and discuss the application or reasons to gather the specific results. To compare the results with other literature to find out exciting insights.

6. PLOS authors have the option to publish the peer review history of their article (what does this mean?). If published, this will include your full peer review and any attached files.

Reviewer #1: No

Reviewer #2: No

---

## [Author Response · Author response to Decision Letter 0]

13 May 2021

Emily Chenette, Editor-in-Chief April 23, 2021

PLOS ONE 

Re: manuscript no. PONE-D-20-40025, “Analysis of Individual Differences in Pelvic and spine alignment in seated posture and Impact on the Seatbelt Kinematics using human body model”

Dear Dr. Emily:

Thank you for providing us with comments by the reviewers regarding our manuscript no. PONE-D-20-40025 R1, “Analysis of Individual Differences in Pelvic and spine alignment in seated posture and Impact on the Seatbelt Kinematics using human body model.” We have answered their comments point by point. We hope that you decide that our revised manuscript should be published in your journal.

Journal Requirement #1

Response #1: I corrected PLOS ONE's style.

Journal Requirement #2. 

Thank you for providing a statement in your Methods section that informed written consent was given by all participants. We ask that you additionally include this statement in the Ethics Statement.

Please amend your Methods section to clarify where the THUMS and OASYS software products can be accessed, and if they are freely available to the public.

Response #2: I state in the Ethics Statement including Requirement #3.

 We purchased the THUMS and OASYS software.

 However, the THUMS is free to use from April, 2021.

Journal Requirement #3. 

Thank you for including your ethics statement: "This study was conducted in collaboration with project partner Mazda Motor Corporation (Aki-gun, Hiroshima, Japan) with approval from the ethical review board of the corresponding author’s host institute.". 

Response #3a: I added to “The research was approved by the ethics committee at the Center for Clinical Research, Yamaguchi University Hospital (Ube, Japan; approval no. H29-052).” in the Ethics Statement.

We are also in the process of submitting a new ethics document that will be used for this simulation using the data from H29-052. If the paper is accepted, we will add the number to it.

Response #3b: I add the same text to the “Ethics Statement”.

Journal Requirement #4. In your Data Availability statement, you have not specified where the minimal data set underlying the results described in your manuscript can be found. PLOS defines a study's minimal data set as the underlying data used to reach the conclusions drawn in the manuscript and any additional data required to replicate the reported study findings in their entirety. All PLOS journals require that the minimal data set be made fully available. For more information about our data policy, please see http://journals.plos.org/plosone/s/data-availability.

Response #4:

 I added manuscript and uproad Supporting Information files. 

The data that support the findings of this study are available from the corresponding author, HY, and YI upon reasonable request and reference number.

Journal Requirement #5. Thank you for stating the following in the Financial Disclosure section:'Mazda Motor Corporation provided $30,000 to cover joint 2017 research expenses for this study.' 

We note that one or more of the authors have an affiliation to the commercial funders of this research study: Mazda Motor Corporation ～～.

Response #5:

I corrected Coverletter.

Norihiro Nishida developed research plan.

Norihiro Nishida and Hiroki Yamagata measured X-ray data.

Norihiro Nishida, Fei Jiang, and Xian Chen analysed FEM using THUMS.

Junji Ohgi, Hidenori Suzuki, Yasuaki Imajo, Masahiro Funaba, and Takashi Sakai played role of guidance and modification of reseach.

Tomohiro Izumiyama, Ryusuke Asahi, Shigeru Sugimoto, and Masanobu Fukushima are not have any additional role in the study design, data collection and analysis, decision to publish, or preparation of the manuscript.

Funding

Mazda Motor Corporation provided $30,000 to cover joint 2017 research expenses for this study.

We have financial relationships to disclose. However, The funder provided support in the research materials (car seat, FEM software, English editing fee, OASYS, and THUMS), but did not have any additional role in the study design, data collection and analysis, decision to publish, or preparation of the manuscript. This does not alter our adherence to PLOS ONE policies on sharing data and materials.

Journal Requirement #6. Please amend either the title on the online submission form (via Edit Submission) or the title in the manuscript so that they are identical.

Response #6: It was changed to the title on the online submission form.

Journal Requirement #7. We note that Figure 1 includes an image of a participant in the study. ～～

　Please amend the methods section and ethics statement of the manuscript to explicitly state that the patient/participant has provided consent for publication: “The individual in this manuscript has given written informed consent (as outlined in PLOS consent form) to publish these case details”.

Response #7: 

 I corrected the methods section and ethics statement. And volunteer of Fig.1 signed informed consent form.

 “This study was conducted in collaboration with project partner Mazda Motor Corporation (Aki-gun, Hiroshima, Japan). The research was approved by the ethics committee at the Center for Clinical Research, Yamaguchi University Hospital (Ube, Japan; approval no. H29-052). Written informed consent for this study and its publication was obtained from all subjects. And, the individual (for example Fig. 1) in this manuscript has given written informed consent (as outlined in PLOS consent form) to publish these case details.”

Reviewer #1

We are grateful to reviewer #1 for their critical comments and useful suggestions that have helped us to improve our paper. As outlined in the responses below, we have taken all reviewer comments and suggestions into account in the revised version of our manuscript.

Specific comments

Comment #1: Line 95: Should it be PA (01, 10, and 50)?

Response #1: Thank you for your pointing out. I corrected “PA (01, 10, and 50)”.

Comment #2: Line 120: Please show more details for the previous study. For example, what variables were involved in Kimpira’s study? How different is the predicted injury than the real injury?

Response #2: Another reviewer pointed out to write the Introduction in detail. 

I corrected Introduction section as follows;

“Although the global death rate from motor vehicle accidents is declining, approximately 1.35 million people are killed each year as a result of road traffic crashes according to the World Health Organization's Road Traffic Crashes database. Whiplash injury can reduce an accident victim’s quality of life for a significant amount of time, ranging from weeks to a few months, with symptoms persisting for years in some cases (1). Various efforts have been made to reduce human injuries from car accidents by improving car seat belts and the car itself, including the materials used in manufacture, as well as by analyzing car accidents. Seat belts have proven to be important in reducing injuries to car occupants during crashes (2). However, deaths and injuries from car accidents are still too frequent and there remains considerable scope for improving car seats and seat belts, especially since there are clinical reports of injury resulting from seat belts during accidents (3-5). These studies do not refer to the mechanism by which seat belts, which are meant to protect humans, can also cause injuries in some circumstances. Actual crash studies using humans are not practical or ethically feasible. Crash analysis using dummy dolls in cars is important, but this is quite expensive and can't be done more than once(6). Computer simulation using the finite element method (FEM) is an important way to analyze injury from car accidents, but so far there are few published reports. Kimpira analyzed injuries from car accidents and used FEM human body models to simulate the safety profile of various seats and seat belts (HBMs) (7). They did this by adapting a previously used male FEM model to a female FEM model. In addition, Danelson et al. performed a pediatric-sized FEM and brain injury analysis for a car crash (8). Davis et al. performed a seated FEM analysis and reported that pelvic fractures were more likely to occur in female models (9). 

However, it is difficult to fully replicate injuries from car accidents. To make the analysis more thorough, we considered the possibility that the occupant's seating alignment was also involved. As a basis for this hypothesis, it is well known that changing from a standing to a sitting position causes changes in lumbar and pelvic alignment (10-17). Individual differences in skeletal and seated alignment may explain why simulations do not fully replicate accidents. 

In the present study we investigated factors that influenced FEM simulation and the effect of angle of the spine and pelvis on the injury to a person in an accident when sitting in a car seat. 

THUMS version 4.0 (THUMS ver. 4.0; TOYOTA MOTOR CORPORATION, Toyota, Aichi, Japan) and the finite element method (FEM) were used to analyze the impact to seat belts and the human body at the time of a collision by modeling the skeletal radiological data of a person sitting in a car seat. A better understanding of human and seat belt movement according to the angle of the spine and pelvis should lead to improved analysis of human injury mechanisms and risk reduction.

”

Comment #3: Line 134-137: This paragraph should be in the Acknowledgement.

Response #3: The journal specified that this paragraph and the Ethic statement should be written within the material and Method. Let it be in this position.

Comment #4: Line 144: Are the seat size and seat height similar to the ones in the Mazda car?

Response #4: The seat is the same as the driver's seat of Mazda3, only the base device was specially made for the X-ray. I have added some text to the Method section.

“Lateral x-ray images of the global spine alignment in the sitting position were acquired with participants in a car seat (same as the driver's side seat of Mazda3).”

Comment #5: Line 183: For all the results, please also provide standard deviations. Although range is useful, standard deviation is considered the more reliable measure for statistical analysis.

Response #5: Thank you for your comment. I added standard deviations.

“The mean values of alignment in the sitting position were as follows: LL, -0.49°±9.58°(range, -26° to 26.6°); SS, -16.5°±8.64° (range, -41° to 4.3°); and PA, -45.57° ±9.80°(range, -67.0° to -20.0°).”

Comment #6: Line 191/Table 1: Please explain the parameters better. Will the change of these parameters result in injury? Can you calculate the injury risk? One aim of this study is to help decrease the risk of injury during an accident. Therefore, the estimated injury probability will be helpful.

Response #6: Thank you for your comment.　

This calculation is estimating the injury risk.

I added Method Section as follows;

　“Using the alignment, we measured the movement and rotation of the body and seatbelt during car crash and analyzed the injury risk.”

 “Using the results of the FEM, the amount of lap-belt cranial sliding-up anterior movement of the pelvis, posterior tilt of the pelvis for seeing abdominal, aorta sand spine injury. Head injury criterion (HIC), second cervical vertebrae (C2) compressive load, C moment for seeing cervical spine injury, chest deflection (upper, middle, and lower) for seeing chest injury and rib injury, left and right femur load, and shoulder belt force for seeing limbs injury were measured.”

I added Result Section as follows;

　“The measurements are shown in Table 1. At the time of the collision, all parameter had increased. Lap-belt cranial sliding-up was 10.8, 20.6, and 25.7mm, anterior movement of pelvis was 139.8, 154.0, and 165.1mm for PA50, PA10, and PA01, posterior tilt of the pelvis was 41.75, 45.05, and 46.78°, HIC was 532.9, 601.4, and 840.9, C2 moment was 1.88, 2.99, 5.11mm, and chest deflection lower was 25.4, 28.7 and 30.5 for PA50, PA10, and PA01. These parameter were gradually increased by from PA50 to PA01. The other parameters did not show a gradual increasing trend with PA.”

and

 “PA50 tended to have less Lap-belt cranial sliding-up, Anterior movement of pelvis, HIC, C2 moment, and lower chest than PA10 and PA01. There were no characteristic differences in the upper/middle chest or lower limbs due to alignment.”

Reviewer #2

We are grateful to reviewer #2 for critical comments and useful suggestions that have helped us to improve our paper. As outlined in the responses below, we have taken all reviewer comments and suggestions into account in the revised version of our manuscript.

Specific comments

Comment #1: 1. The introduction is too short. The authors need to identify the contribution, novelty, and related works comprehensively.

Response #1: I corrected Introduction section as follows;

“Although the global death rate from motor vehicle accidents is declining, approximately 1.35 million people are killed each year as a result of road traffic crashes according to the World Health Organization's Road Traffic Crashes database. Whiplash injury can reduce an accident victim’s quality of life for a significant amount of time, ranging from weeks to a few months, with symptoms persisting for years in some cases (1). Various efforts have been made to reduce human injuries from car accidents by improving car seat belts and the car itself, including the materials used in manufacture, as well as by analyzing car accidents. Seat belts have proven to be important in reducing injuries to car occupants during crashes (2). However, deaths and injuries from car accidents are still too frequent and there remains considerable scope for improving car seats and seat belts, especially since there are clinical reports of injury resulting from seat belts during accidents (3-5). These studies do not refer to the mechanism by which seat belts, which are meant to protect humans, can also cause injuries in some circumstances. Actual crash studies using humans are not practical or ethically feasible. Crash analysis using dummy dolls in cars is important, but this is quite expensive and can't be done more than once(6). Computer simulation using the finite element method (FEM) is an important way to analyze injury from car accidents, but so far there are few published reports. Kimpira analyzed injuries from car accidents and used FEM human body models to simulate the safety profile of various seats and seat belts (HBMs) (7). They did this by adapting a previously used male FEM model to a female FEM model. In addition, Danelson et al. performed a pediatric-sized FEM and brain injury analysis for a car crash (8). Davis et al. performed a seated FEM analysis and reported that pelvic fractures were more likely to occur in female models (9). 

However, it is difficult to fully replicate injuries from car accidents. To make the analysis more thorough, we considered the possibility that the occupant's seating alignment was also involved. As a basis for this hypothesis, it is well known that changing from a standing to a sitting position causes changes in lumbar and pelvic alignment (10-17). Individual differences in skeletal and seated alignment may explain why simulations do not fully replicate accidents. 

In the present study we investigated factors that influenced FEM simulation and the effect of angle of the spine and pelvis on the injury to a person in an accident when sitting in a car seat. 

THUMS version 4.0 (THUMS ver. 4.0; TOYOTA MOTOR CORPORATION, Toyota, Aichi, Japan) and the finite element method (FEM) were used to analyze the impact to seat belts and the human body at the time of a collision by modeling the skeletal radiological data of a person sitting in a car seat. A better understanding of human and seat belt movement according to the angle of the spine and pelvis should lead to improved analysis of human injury mechanisms and risk reduction.”

Comment #2: Methods- “The seat’s surface and reclining angles were fixed.” Due to the X-ray was used to perform once to collect the data, how to ensure the sitting posture is correct or the same as usual when the participants driving? Does the reclining angle of the seat affect the lumbar angle and pelvis?

Response #2: Important question. Reclining was also performed in the pre-experimental trial. However, the reclined car seat could not fit into the X-ray machine. In addition, LL of some participants did not include within the imaging range. Therefore, the seat angle had to be fixed to allow all subjects to capture the entire spine in the X-ray machine. The seat angle had to be fixed in order to capture the whole spine in the X-ray machine. Although we envisioned a situation where the drivers would change positions while driving, we could not do so because the experiment would take too long. The limitation of this experiment is that the initial position was taken, which is different from the original posture of the subject when driving.

Comment #3: The FE analysis needs emphasis clearly. How to measure each measure (LL, SS…)? Not only provided the definitions but also the detail of collection and measurements.

Response #3: I corrected Methods and Result section.

“The human model for collision analysis, THUMS ver. 4.0 AM50, was used. Alignment of THUMS was modified using pre-processing software Oasys PRIMERTM (ARUP, London, UK). We purchased both software. 

Since analyzing each subject's data have resulted in a huge amount of analysis, we focused on PA. In order to select representative values of PA, normal distribution and histogram were used for the analysis of PA (Fig. 3). The results showed that the 1st percentile of the PA was 67°, the 10th percentile was 57°, and the 50th percentile was 45°. The LL angles of the volunteers for each PA angle were -8°, -8°, and 27°. In addition, the volunteer SS angles for each PA angle were -20°, -10°, and 5°. In order to create a model corresponding to those angles, the skeletal alignment of the THUMS model was modified by giving a Prescribed-Motion (enforced displacement) on the vertebrae of the thoracic spine, lumbar spine and pelvis. Femur joint was allowed to rotate but not translate. The hands, lower extremities and cervical vertebrae were completely constrained.

Comment #4: Lacks of statistical analysis results to support the findings.

Response #4: With regard to this comment, please note that we performed a statistical analysis of seated X-rays in a previous paper. 

Nishida N, Izumiyama T, Asahi R, Iwanaga H, Yamagata H, Mihara A, Nakashima D, Imajo Y, Suzuki H, Funaba M, Sugimoto S, Fukushima M, Sakai T. Changes in the global spine alignment in the sitting position in an automobile. Spine J. 2020 Apr;20(4):614-620.

For the present research, a crash computer simulation with different angles of lumbar vertebrae and pelvis was performed. Conditions were identical except for the angle. Because it is a simulation, the results are the same no matter how many times it is done. 

The following sentence has been included in the Limitation section to address this comment:

“With computer simulation, the results are the same regardless of how many times the experiment is repeated and hence statistical analysis was not performed.”

Comment #5: The discussion is weak. Authors need to focus on the results of the study and discuss the application or reasons to gather the specific results. To compare the results with other literature to find out exciting insights.

Response #5: 

I corrected Discussion as follows;

“This study investigated alignment of the human body when sitting in a car seat; thus, simulating human damage patterns of accidents and investigating seat belt and seat positions. One of the aims was to analyze car crashes using a human model and help decrease the risk of injury during an accident. The study also provides evidence for the development of other vehicle seats and human simulation analysis.

Several computer simulation analyses using THUMS and FEM for automobile accidents have been reported. Kimpara et al. analyzed a female model and compared the results with a male model(21). Andersson et al. performed an analysis on child size and reported that light vehicles required higher performance interior restraint systems than larger vehicles(22). Danelson et al. and Digges et al. analyzed lung contusions during motor vehicle accidents (23, 24). Klein et al. reported that body shape altered impact to the femur (25). Mattos et al. analyzed the damage to the head and cervical spine during a rollover (26). Jones et al. performed an analysis of lumbar spine fractures at the time of the collision (27). Paas et al. analyzed the mechanism of shoulder injury at the time of an accident (28). Xiao et al investigated of chest injury mechanism of seat belt using FEM(29). Our analysis also showed movement of shoulder, chest, and cervical spine, as previously reported. These analyses can be viewed as useful research for the development of human models and improving vehicle safety. However, there are challenges resulting from collision conditions and model validation, and there was no analysis focusing on vertebral angle.　

There were some reports of individual differences in the angle of lumbar and pelvis with sitting position. Studies by Suzuki et al. and Endo et al. examined the alignment in standing and sitting positions in middle-aged and elderly individuals, and in young adults, respectively (12, 13). Lee et al. reported that lumbar spine lordosis and pelvic tilt were decreased when study participants were sitting (14). Nishida et al. reported that body alignment changes with the car seat (17). However, there were no reports examining the effects of lumbar spine or pelvic alignment in the analysis of car accidents. In the present analysis, three models were created with different angles of the pelvis and lumbar vertebrae, and it was found that there were large individual differences in the angle changes. In the present analysis, we focused on these clinical reports.

There have been several reports on clinical injuries caused by traffic accidents. It has been clinically identified that there is an increased whiplash injury risk when the occupants head is rotated or inclined prior to the impact (30, 31). Radanov et al reported a higher incidence of rotated or inclined head position at the time of impact was correlated with increased severity of symptoms such as neck pain, shoulder pain, and headache (32). In the present analysis, C2 compressive load also increased during the crash, and HIC and the C2 moment was exacerbated by the larger PA. The present analysis also showed that the accident added movement to head and the cervical spine.

Herath reported a shearing transection of a gastroduodenal junction caused by an inappropriate seat belt (33). Muraoka et al reported that uterine trauma and Intrauterine fetal death caused by seatbelt injury(34).Tomic et al reported that seat-belt abdominal aortic injury and rib fracture(35). Ramachandra et al reported on seat belt-induced increase in abdominal pressure during an accident(36). Abbas et al. reported that seatbelt-related injuries include spinal, abdominal or pelvic injuries and the presence of a seatbelt sign must raise the suspicion of an intra-abdominal injury due to seatbelt repositioning during traffic accidents. The presence of a seatbelt sign must raise the suspicion of an intra-abdominal injury(37). In the present study, actual collision analysis was carried out in three human models based on the measured data of spine and pelvis angles. Lap-belt cranial slide-up, anterior movement of the pelvis, and posterior tilt of the pelvis moved in all alignment conditions, suggesting that traffic accidents can cause abdominal injuries, aortic injuries, and spinal fractures. Some reseacher said inappropriate seat belt use, but we found that increasing LL and decreasing PA allowed for better seat belt application to the iliac bone and better control of human movement. It has the potential to reduce injury in the event of an accident. Jiang et al. reported a higher risk of seat belt injury when the seat belt was above the ASIS, and the present study also found that as pelvis tilt decreases, the seat belt position shifted to the head side and the ability to control the body decreased (38). This study found that the spinal alignment, pelvic tilt and seatbelt position may decrease/increase the risk of injury.

There are several limitations to our study. The size and imaging range of the x-ray equipment did not allow frontal imaging in the sitting position and only allowed lateral imaging at one angle (not adjustable by reclining). In addition, we studied only one tissue strength of bones and organs in THUMS; therefore, age-related changes, such as osteoporosis, were not taken into account. In addition, the soft tissue thickness of the thighs and abdomen, and variations in the size of the pelvis were not included in this model. With computer simulation, the results are the same regardless of how many times the experiment is repeated and hence statistical analysis was not performed. These are issues to be addressed in the future.

Despite these limitations, an analysis of a hypothetical car accident using a human model with altered vertebral angle and PA showed changes in seat belt position and damage to the human model. This will be a useful reference for future accident analysis and research on the appropriate seat shape and seat belt position.”

Sincerely,

The authors of the manuscript “Analysis of Individual Differences in Pelvic and spine alignment in seated posture and Impact on the Seatbelt Kinematics using human body modeｌ”

---

## [Decision Letter · Decision Letter 1]

21 Jun 2021

Analysis of Individual Differences in Pelvic and spine alignment in seated posture and Impact on the Seatbelt Kinematics using human body model

PONE-D-20-40025R1

Dear Dr. Nishida,

We’re pleased to inform you that your manuscript has been judged scientifically suitable for publication and will be formally accepted for publication once it meets all outstanding technical requirements.

Kind regards,

YunJu Lee

Academic Editor

PLOS ONE

Additional Editor Comments (optional):

Reviewers' comments:

Reviewer's Responses to Questions

**Comments to the Author**

1. If the authors have adequately addressed your comments raised in a previous round of review and you feel that this manuscript is now acceptable for publication, you may indicate that here to bypass the “Comments to the Author” section, enter your conflict of interest statement in the “Confidential to Editor” section, and submit your "Accept" recommendation.

Reviewer #1: All comments have been addressed

Reviewer #2: All comments have been addressed

2. Is the manuscript technically sound, and do the data support the conclusions?

Reviewer #1: Yes

Reviewer #2: Yes

3. Has the statistical analysis been performed appropriately and rigorously? 

Reviewer #1: Yes

Reviewer #2: Yes

4. Have the authors made all data underlying the findings in their manuscript fully available?

Reviewer #1: Yes

Reviewer #2: Yes

5. Is the manuscript presented in an intelligible fashion and written in standard English?

Reviewer #1: Yes

Reviewer #2: Yes

6. Review Comments to the Author

Reviewer #1: The quality of the paper was improved. The submission presents contain an interesting contribution in terms of the analysis of pelvic and spine alignment in seated posture. This study may help reduce the impact of accidents, and to reconsider the safe seat and seatbelt position in the future.

Reviewer #2: All of my questions are well addressed. The quality of the manuscript is improved. However, there is a minor concern that should be revised.

For the response of the previous Q2, the practical situation that the authors mentioned is not entirely convinced, but it is acceptable. At least, I strongly recommend stating the phenomenon in the limitation.

7. PLOS authors have the option to publish the peer review history of their article (what does this mean?). If published, this will include your full peer review and any attached files.

Reviewer #1: No

Reviewer #2: No

---

## [Editor Report · Acceptance letter]

1 Jul 2021

PONE-D-20-40025R1 

Analysis of Individual Differences in Pelvic and spine alignment in seated posture and Impact on the Seatbelt Kinematics using human body model 

Dear Dr. Nishida:

I'm pleased to inform you that your manuscript has been deemed suitable for publication in PLOS ONE. Congratulations! Your manuscript is now with our production department. 

Kind regards, 

on behalf of

Dr. YunJu Lee 

Academic Editor

PLOS ONE